# A Multilevel Analysis of Risk and Protective Factors for Adolescent Childbearing in Malawi

Jesman Chintsanya [1,*] , Monica Magadi [2] and Gloria Likupe [3]

1 Department of Population Studies, Chancellor College, University of Malawi, P.O. Box 280, Zomba, Malawi
2 Department of Criminology and Sociology, Faculty of Arts Cultures and Education, University of Hull, Hull HU6 7RX, UK; m.magadi@hull.ac.uk
3 Department of Nursing, Faculty of Health Sciences, University of Hull, Hull HU6 7RX, UK; g.likupe@hull.ac.uk
* Correspondence: jchintsanya@cc.ac.mw

**Abstract:** Although teenage pregnancy and childbearing has declined throughout sub-Saharan Africa, the recent increase in teenage pregnancy in countries such as Malawi has prompted interest from social researchers. Using Malawi Demographic and Health Survey (MDHS) data from 2004 to 2015, this study employs multilevel logistic regression to examine the magnitude of change over time in risk and protective factors for teenage childbearing. During this period, teenage childbearing declined from 36.1% (C.I.: 31.5–36.7) in 2004 to 25.6% (C.I.: 24.0–27.3) in 2010 before increasing to 29.0% (C.I.: 27.4–30.7) in 2015. Age and being married (compared to never married) were consistently significantly associated with increased odds of teenage childbearing. However, delaying sexual debut, attaining secondary education, belonging to the richest quintile and rural residence offered protective effects against early motherhood, while Muslim affiliation (compared to Christian denominations) was associated with increased likelihood of teenage childbearing among adolescents. Teenage childbearing remains high in the country, largely influenced by adolescents' early sexual debut and child marriage—risk factors that have hardly changed over time. While individual socioeconomic predictors are useful in explaining the apparent high risk of adolescent fertility among specific subgroups in Malawi, sustained declines in teenage childbearing were not evident at district level.

**Keywords:** teenage childbearing; adolescent; sexual reproductive health; multilevel modelling; Malawi





## 1. Introduction

Global trends estimate that 12 million adolescents between the ages of 15 and 19 give birth every year, and more than 90% of these live births occur in low- and middle-income-countries (WHO 2020). Although the global adolescent birth rate declined from 65 births per 1000 women in 1990 to 44 births per 1000 women in 2018 (SDGS n.d.), sub-Saharan Africa (SSA) still has the highest teenage fertility rates in the world with an estimated 101 births per 1000 women aged 15–19—more than double that of developed countries (UNFPA 2013). Every year, about 3 million girls aged 15–19 years have unsafe abortions, resulting in an estimated 2500 adolescent deaths (Temmerman 2017); an estimated 70,000 teenage girls die each year during pregnancy and childbirth and more than one million infants born to adolescent girls die before their first birthday (WHO 2016). Despite efforts pressing for the improvement of adolescent sexual and reproductive health (ASRH), there is limited understanding of risk and protective factors influencing teenage pregnancies.

Studies have shown that puberty is occurring at an earlier age leading to adolescents' early initiation of intercourse and consequently early marriage, which in turn is positively associated with teen birth (Mensch et al. 2005; Morris and Rushwan 2015). It has been argued that the timing of sexual activity is important in the likelihood of teenage pregnancy especially for early-maturing girls as it puts them at higher risk of forming opposite-sex

relationships. In turn, early sexual activity increases their risks to pregnancy, especially when intercourse occurs without use of contraceptives. Early marriages may also occur as one way of preventing societal shame. Parents would want to marry off young girls to avoid parental shame if pregnancies occur before marriage. In this case, early marriage may be perceived as an avenue of improving teens' social status, ensuring that adolescents obtain respect from the community and independence (Pot 2019). Adolescents who are married or are cohabiting are likely to be pregnant since they are at risk of higher coital frequency which correlates positively with the probability of teen childbearing (Azevedo et al. 2012; Bongaarts and Potter 2013).

Use of modern contraceptives among adolescent girls has generally been known to be effective in so many ways. Notably, the use of contraceptives delays childbearing and prevents unwanted pregnancies among adolescent girls while reducing complications associated with early childbearing (Mensch et al. 2005). The use of contraceptives also ensures that births are well spaced while allowing for a longer breastfeeding period which is beneficial for the child's and the mother's health (Neal et al. 2012). However, recent findings suggest that adolescents' use of contraceptives has varied significantly in the SSA region, and that half of adolescent women do not use modern contraceptive methods (Sully et al. 2019). This indicates that there is a significant gap between teen sexual activity and exposure to knowledge of use of contraceptives, often attributed to misconceptions and prescribed religious and cultural norms. Consequently, adolescents have unwanted births which sometimes lead to unsafe abortions and complications (Magadi 2006; Gunawardena et al. 2019). Moreover, if early childbearing within marriage is socially accepted and even encouraged, efforts aimed at increasing contraceptive use among adolescents may be attenuated (Eloundou-Enyegue and Stokes 2004; Acharya et al. 2010; McQueston et al. 2012).

Such proximate factors may be mediated by cultural and socioeconomic environments which are strongly linked to adolescents' access to sexual and reproductive health (SRH) information. Depending on the norms of the society, the environment adolescents find themselves in may act as an agent that predisposes them to the likelihood of childbearing (Poudel et al. 2018; Gunawardena et al. 2019). Brown et al. (2002) hypothesize that urban areas may be responsible for high adolescent fertility because adolescents who are separated from parents are deprived of daily access to useful models and sources of advice and are therefore likely to be involved in mischievous behaviour.

Furthermore, a systematic review which involved pooled meta-analysis showed that adolescents (15–19 years) who reside in 20% of poorest households in rural areas face extra vulnerability associated with being young, compared to counterparts in the urban areas (Kassa et al. 2018). In addition, adolescent girls living in rural areas have twice as many births as those in urban areas (Eloundou-Enyegue and Stokes 2004). The negative health outcomes are in part driven by poor socioeconomic environments in the rural areas. Adolescents in the rural areas have limited access to information about contraceptives. Some reside in hard-to-reach areas and may lack appropriate and adequate services to their SRH needs (Obasohan 2015; Magadi 2006). Consequently, these poor socioeconomic environments accelerate the likelihood of unprotected premarital sex and teenage pregnancy among young women in rural areas than among those in the urban areas (Eloundou-Enyegue and Stokes 2004).

Some studies have shown that religion may be linked to teenage childbearing. Some religious teachings and morals may prohibit the use of contraception (Obasohan 2015) or may lead to inconsistent or incorrect use of contraceptives by adolescents (Yeatman and Trinitapoli 2008). Religion may also prescribe what to teach adolescents about sexuality and the prevention of teen pregnancy—often focusing on abstinence before marriage (Heaton 2011; Pot 2019; Munthali et al. 2006)—while excluding the teaching of contraceptives (Strayhorn and Strayhorn 2009). This may result in adolescents having incorrect and inconsistent use of contraception resulting in unintended pregnancies.

Societal norms—being an indirect determinant of adolescent fertility—affect contraceptive service access in the sense that they may create biases with regard to how adolescents

access sexual and reproductive health (SRH) services (Likupe et al. 2020). For example, adolescents may shun the services because of the providers' judgmental attitude toward unmarried adolescents (Kennedy et al. 2013; Chilinda et al. 2014; Rosenberg et al. 2018).

Studies reveal that pregnancy and childbirth are riskier for young mothers than for adult women (Morris and Rushwan 2015; Ngome and Odimegwu 2014). Due to decreased likelihood of adequate prenatal care and nutrition, it has been found that teen mothers are up to three times more likely to die from pregnancy-related causes than adult mothers, and they are twice as likely to suffer from severe complications during pregnancy and birth (Neal et al. 2012). Teen mothers are also more likely to indulge in unsafe abortion than adult mothers (Gunawardena et al. 2019).

Teen childbearing can gravely affect teen mothers' adult life both socially and economically. With regard to education, teen mothers are likely to drop out of school and their children rarely access education, resulting into a vicious cycle of intergenerational poverty. Those who continue with school face frequent interruptions and absenteeism leading to poor academic results (Chiavegatto Filho and Kawachi 2015; Rosenberg et al. 2018). Consequently, they achieve fewer desirable competencies and skills, which lowers their employment prospects in the long run (McQueston et al. 2012; Akella and Jordan 2015).

However, it has been argued that this relationship is problematic to bear an impact because often times adolescent girls may have already been underperforming in school and the duration of schooling is short (Rosenberg et al. 2018). Additionally, where norms allow early marriage and childbearing for girls, adolescents' childbearing may have little influence in school dropout (Eloundou-Enyegue and Stokes 2004; Rosenberg et al. 2018).

The burden associated with teenage pregnancy comes with a large social toll not only on the individual woman but also on the children and family. Given that in many cases the teen mothers live in the parental households after the birth of the child, both the parents and siblings are likely to be affected (McQueston et al. 2012; Poudel et al. 2018). A study in South America which compared the cost of childbearing of teen mothers to adolescents who delayed childbearing found significant negative effects on the prospects for marriage of teen mothers (Arceo-Gómez and Campos-Vazque 2014).

In Malawi, teenage childbearing generally declined between 1992 (35%) and 2010 (26%), reaching an all-time low level of 121.0 conceptions per 1000 women aged 15–19 in 2010 (NSO and ICF 2017). However, the teen childbearing rate went up in 2015 (29%). This presents a major concern as the contribution of adolescents to total births among women aged 15–49 is reported to be the highest in the east and central sub-Saharan Africa, with a share of 16%, which is notably higher than other countries in the region such as Zimbabwe (12%), Tanzania (12%), Ethiopia (8%) and Kenya (11%) (UNDESA 2017).

Similarly, early sexual activity is higher in Malawi than in some countries in the region. In Malawi, 20% of girls reported having had first sexual intercourse by the exact age of 15 compared to Kenya (15%), South Africa (7%), Tanzania (13.6%), Zambia (17.1%) and Zimbabwe (5.7%) (NSO and ICF 2017). While the 2015–16 Malawi Demographic and Health Survey (MDHS) shows that male adolescents (22.8%) aged 15–19 engage in their first sexual intercourse much earlier than girls (12.8%), girls get married much earlier: 26.8% of young adolescent girls compared to less than 3.4% of boys are married before the age of 15 (NSO and ICF 2017).

Faced with these challenges and as part of its national development agenda to reduce adolescent fertility, Malawi, through the Family Planning 2020 initiative, committed to increasing the prevalence rate of modern contraceptives from the baseline of 38% in 2012 to 60% for all women by 2020, with a particular focus on the 15–24-year age group (Family Planning 2020 2019). Some of the major strategies include: raising the legal marriageable age from 16 to 18 years old (MoEST and UNICEF 2017); increasing youth-friendly health service (YFHS) centres where adolescents and the youth can obtain SRH services; increasing demand for family planning method uptake by promoting youth outreach and static health services; youth community mobilisation of health practices; and having role models in the rural areas that advocate against early marriages (Youth Friendly Health Service (YFHS)

Evaluation Study 2014). Furthermore, intervention programs have markedly increased. For example, comprehensive sexuality education has been introduced in public schools to increase learners' knowledge of the ways of preventing pregnancy and HIV (YFHS 2014). Also introduced is the reintegration policy to support new mothers' re-entry into school and provide them with emotional and financial support. The policy also aims to prevent child marriage and further early pregnancies (Chalasani et al. 2012).

The foregoing context warrants isolating individual and contextual factors that mediate adolescent childbearing and district differences. Understanding how individual and contextual factors influence teenage pregnancy may enable the identification of risk areas and appropriate planning of intervention and strategies to reduce teenage pregnancy. Although there has been research focusing on the determinants of teenage childbearing in the region, studies that focus on Malawi are scarce. To the authors' knowledge, only a handful of studies have attempted to address certain aspects of risk factors affecting teenage pregnancy in Malawi, using standard regression analysis (Palamuleni 2017) and a decomposition approach (Chirwa et al. 2019). Overall, very few studies that examine both individual and contextual factors have directly addressed the problem of teenage pregnancy. In such studies, Malawi was part of multicountry studies (Wado et al. 2019).

Thus, incorporating a multilevel approach allows simultaneous investigation of the effects of individual and group level risk factors. The present study aims to fill this gap by: (i) examining the pattern of change in individual and contextual risk factors of teenage pregnancy leading to the disruption of declining trend; and (ii) acknowledging that individuals within districts may have some degree of correlation due to unobserved common characteristics, which ultimately, may result in incorrect conclusions on the effects of associated factors (Hox et al. 2018; Rabe-Hesketh and Skrondal 2012).

The specific objectives of this study were to: (i) explore associations related to risk and protective factors of teenage childbearing in Malawi at both individual and district levels; (ii) examine, within a multilevel regression framework, the magnitude of change over time in the factors associated with teenage childbearing, and (iii) provide policy implications. In doing so, findings from the study may suggest a future direction for interventions that seek to mitigate the adverse consequences of teenage childbearing in Malawi and similar SSA countries.

## 2. Methods

### 2.1. Data Sources

This paper is based on a secondary analysis of Malawi Demographic and Health Survey (hereafter MDHS). The analyses are restricted to 15–19 year old female adolescent sample obtained from the 2004, 2010 and 2015–16 MDHS. The data were downloaded with permission from Measure DHS. DHS surveys are designed to assist developing countries in monitoring indicators for development which include health, nutrition programs, fertility and mortality awareness, as well as behaviour change regarding HIV and AIDS and other sexually transmitted infections (Rutstein and Rojas 2006).

The MDHS uses a two-stage stratified cluster sampling technique, which allocates census enumeration areas (EAs) as sampling units for the first stage. EAs are further subdivided into the three regions (i.e., north, central and southern regions) and the 28 districts of Malawi. An EA (i.e., cluster) is a geographic area consisting of a convenient number of dwelling units which serve as a counting unit in a census. The EAs were stratified based on whether they were urban or rural. All households in the selected cluster were listed. Households comprised the second stage of sampling selected with an equal probability systematic selection per enumeration area (NSO and ICF 2017). All eligible respondents from selected households were invited to participate in the MDHS. In each of 2004, 2010 and 2015–16 MDHS surveys, out of 12,229, 23,748 and 25,146 eligible women respondents of reproductive age (15–49 years), the response rates were 96.9%, 98% and 97.7%, respectively. The current study focuses on MDHS subsample, comprising female

adolescents aged 15–19 years: N = 2407 (2004 MDHS); N = 5040 (2010 MDHS) and N = 5273 (2015–16 MDHS).

## 2.2. Outcome Variable

The study outcome was a binary variable, which took a value of one (1) if women aged 15–19 years reported a live birth or were pregnant with first child at the time of interview, or a value of zero (0) if not (Magadi 2017; Poudel et al. 2018; Chirwa et al. 2019).

## 2.3. Explanatory Variables

Table 1 presents the demographic, socioeconomic and contextual explanatory variables, which pertain to women's characteristics. The variables were obtained from the various sections in the women's MDHS questionnaire. The selection of these explanatory variables was based on perceived predictors of teenage childbearing, largely informed by existing literature included in the introduction on factors associated with teenage childbearing.

**Table 1.** Description of independent variables.

| Explanatory Variable | Description and Coding of the Variable |
| --- | --- |
| Demographic | |
| Age of a respondent | Categorised into (1) 15, (2) 16, (3) 17, (4) 18 and (5) 19. |
| Early sexual debut [a] | Coded as 1 if adolescent engaged in sexual intercourse before the age of 15 completed years, 0 if otherwise |
| Marital status [b] | Coded (1) never married and (2) ever married. |
| Sociocultural and economic variables | |
| Education | The highest educational level attained by an individual: (0) no education, (1) primary—1–4, (2) primary—5–8, (3) secondary and higher. |
| Ever used modern contraceptives | Adolescents were categorised into (1) if they ever used modern contraceptives or (0) if otherwise. |
| Religion | Religion was grouped into (1) Catholic, (2) Presbyterians, (3) Pentecost, (4) Muslim and (5) other. |
| Ethnicity | Ethnicity in Malawi has five major stratifications grouped as (1) Chewa, (2) Ngoni, (3) Yao, (4) Lomwe and (5) other |
| Wealth Index [c] | Obtained using principal component analysis (PCA) following the standard methodology and divided into five equal groups of 20% of household quintiles (poorest, poorer, medium, richer and richest) at the national level |
| Source of family planning message | Coded from whether or not an individual had heard family planning messages on the radio, on television or read family planning messages in the newspaper or watched. A "yes" response was coded (1) and (0), if otherwise. |
| Occupation | Recoded as (1) currently employed or (0) not currently working |
| Contextual factors | |
| Place of residence | Place of residence was categorised into (1) urban and (2) rural. |
| District | (1) Chitipa, (2) Karonga, (3) Nkhatabay, (4) Rumphi (5) Mzimba, (6) Likoma, (7) Kasungu (8) Nkhotakota, (9) Ntchisi, (10) Dowa, (11) Salima, (12) Lilongwe, (13) Mchinji (14) Dedza (15) Ntcheu, (16) Mangochi, (17) Machinga, (18). Zomba (19) Chiradzulu, (20) Blantyre, (21) Mwanza (22) Thyolo, (23) Mulanje, (24) Phalombe, (25) Chikwawa, (26) Nsanje, (27) Balaka and (28) Neno |

Notes: [a] (Chirwa et al. 2019; Magadi 2017); [b] Too few cases to be analysed as separate categories of divorced and separated. [c] See (NSO and ICF 2017).

The *svy* Stata command was applied to sampling weights for frequencies and proportions in bivariate analyses to account for nonproportional allocation of the sample to strata (urban and rural dwellings) and regions (NSO and ICF 2017). Bivariate statistical analyses based on cross tabulations with chi-squared tests were used to assess the relationship between each of the demographic, socioeconomic and contextual factors associated with

teenage childbearing with values of $p < 0.05$ taken as significant. Furthermore, the analyses accounted for the clustering effect of the survey design, in which teen mothers were nested within districts.

### 2.4. Multilevel Modelling

Multilevel regression analysis is deemed necessary for data with a nested structure like that of MDHS, where individual observations have some degree of correlation within a district because of common characteristics they share. When potential correlation within the upper level is ignored and only the individual level characteristics are considered, it might lead to a violation of the assumption of independence between observations. By using the clustering information, it enables us to obtain statistically efficient estimates of regression coefficients (Rabe-Hesketh and Skrondal 2012; Hox et al. 2018). Therefore, to obtain the mixed effect (fixed effect for both the individual- and community-level factors and a random effect for the between district variation), a two-level mixed-effect logistic regression analysis was used in this study. Thus, the log of the probability of teenage childbearing was modelled in the following form:

$$\log\left[\frac{\pi_{ij}}{1 - \pi_{ij}}\right] = \beta_0 + \beta_1 X_{ij} + \beta_2 Z_{ij} + u_j$$

where $\log[\pi ij/(1 - \pi ij)]$ is the logit of $\pi\_ij$ which is the probability of an adolescent $i$ in district $j$ experiencing childbearing; $X_{ij}$ and $Z_{ij}$ are vectors of individual-level and district-level characteristics; $\beta_0$, $\beta_1$ and $\beta_2$ are corresponding parameter estimates; and $u_j$ is the random effect at district level (Hox et al. 2018). As adolescent childbearing could be affected by district variation, and to test for a broader cultural influence on teenage childbearing, each of the 28 districts of Malawi was included in the second level of the model for each survey year.

### Model Specification

For each year, separate regression analyses were conducted, taking into account certain variables that are closer (proximate determinants) to influencing the outcome variable while others are distal (socioeconomic factors) and examined how they vary over the study period. This ensured that the contribution of each variable was detected, and its mediation influence was assessed. The process followed a four-stage modelling process: first, running the null model that included only the district as the independent variable (Model 1). Then, demographic variables, i.e., age of adolescent, early sex debut and marital status, were added (Model 2). Following this, the model examined the likelihood of adolescents experiencing teenage childbearing by incorporating socioeconomic factors; education, use of modern contraceptives, employment, wealth index, source of family planning messages, religion, ethnicity and place of residence (Model 3). The last step (Model 4) included all the demographic and socioeconomic factors.

For each model, parameters including the deviance of the multilevel model ($-2$*loglikelihood), badness-of-fit statistics were calculated, where lower values indicate that the model has a better fit to the data, while also estimating the interdependency between the districts using $p_u = \sigma_{u0}^2 / (\sigma_{u0}^2 + \sigma_e^2)$ (Rabe-Hesketh and Skrondal 2012), where $\sigma_u^2$ is the district-level variance and $\sigma_e^2$ is the individual-level variance. An additional correlation test based on the median odds ratio (MOR) was used, defined as the median value of the odds ratio between the area at highest risk and the area at lowest risk when randomly picking out two areas (Twisk 2006). Due to a high number of predictors in the models, multicollinearity tests for each independent measure were completed using multiple regression analysis in Stata. Additionally, bivariate correlations between the predictors were examined. These analyses did not provide any evidence of problems associated with collinearity among predictors.

## 3. Results

### 3.1. Descriptive Results

The univariate summary statistics for adolescents in Table 2 indicate that overall, between 2004 and 2015, there was an increase in the level of education among adolescents aged 15–19. The percentage of adolescents who had no education decreased from 5.5% (2004) to 2.7% (2015), while the percentage of adolescents that had achieved secondary and higher education increased from 18.2% (2004) to 21.7% (2015). The median age at first sex intercourse has marginally changed over time; from 16.6 years in 2004 to 16.4 years in 2015. Similarly, the proportion of adolescents engaging in sex before age 15 reduced from 28.4% in 2004 to 22.7% in 2015–16. There were reduced proportions of adolescents who reported using radio and newspaper as the source of family planning messages. The period experienced an increase in proportion among those who used television as the source of family planning messages. With regard to use of modern contraceptives, the results also show that the proportion doubled from 7.6% in 2004 to 15.2% in 2015.

**Table 2.** Summary statistics for the study sample.

| Background Characteristics | Malawi Demographic and Health Surveys (MDHS) | | | | | |
| | 2004 MDHS | | 2010 MDHS | | 2015–16 MDHS | |
| | Percent | n | Percent | n | Percent | n |
|---|---|---|---|---|---|---|
| Age (years) | | | | | | |
| 15 | 18.6 | 451 | 24.7 | 1246 | 23.8 | 1258 |
| 16 | 19.5 | 474 | 23.0 | 1171 | 17.9 | 971 |
| 17 | 17.8 | 420 | 18.5 | 944 | 18.4 | 941 |
| 18 | 23.2 | 578 | 18.1 | 883 | 20.4 | 1085 |
| 19 | 20.9 | 484 | 15.7 | 796 | 19.6 | 1018 |
| Sexual debut | | | | | | |
| Median (years) [a] | 16.6 | | 16.4 | | 16.4 | |
| Early sex debut | 28.4 | 684 | 23.0 | 1159 | 22.7 | 1197 |
| Marital status | | | | | | |
| Proportion ever married | 36.3 | 903 | 26.2 | 1318 | 26.8 | 1364 |
| Highest education | | | | | | |
| No education | 5.5 | 134 | 3.3 | 140 | 2.7 | 126 |
| Primary 1–4 | 24.2 | 611 | 21.2 | 998 | 17.3 | 816 |
| Primary 5–8 | 52.1 | 1244 | 53.9 | 2843 | 58.2 | 3101 |
| Secondary and higher | 18.2 | 418 | 21.6 | 1059 | 21.7 | 1230 |
| Ever employment | 37.1 | 930 | 36.6 | 1932 | 40.0 | 1291 |
| Wealth index | | | | | | |
| Poorest | 16.5 | 423 | 17.8 | 957 | 18.3 | 863 |
| Poorer | 17.2 | 411 | 17.8 | 918 | 19.1 | 950 |
| Middle | 18.6 | 457 | 19.7 | 1004 | 19.9 | 979 |
| Richer | 21.4 | 537 | 19.7 | 1079 | 19.3 | 1071 |
| Richest | 26.3 | 579 | 25.1 | 1082 | 23.4 | 1410 |
| None | 39.5 | 920 | 47.6 | 2321 | 64.25 | 3300 |
| Radio | 42.3 | 1050 | 31.3 | 1685 | 21.22 | 1163 |
| Television | 2.7 | 64 | 4.2 | 196 | 5.33 | 290 |
| Newspaper | 15.5 | 373 | 16.9 | 838 | 9.2 | 520 |
| Modern contraceptives use | 7.6 | 185 | 9.0 | 461 | 15.2 | 802 |
| Ethnicity | | | | | | |
| Chewa | 34.2 | 757 | 34.8 | 1502 | 36.0 | 1646 |
| Tumbuka | 15.6 | 385 | 12.8 | 838 | 12.4 | 841 |
| Ngoni | 26.2 | 645 | 28.9 | 1490 | 30.8 | 1624 |
| Yao | 12.7 | 368 | 12.6 | 503 | 12.9 | 588 |
| Other | 11.3 | 252 | 10.8 | 707 | 7.9 | 574 |

**Table 2.** *Cont.*

| Background Characteristics | Malawi Demographic and Health Surveys (MDHS) | | | | | |
| | 2004 MDHS | | 2010 MDHS | | 2015–16 MDHS | |
| | Percent | n | Percent | n | Percent | n |
|---|---|---|---|---|---|---|
| Religion | | | | | | |
| Catholic | 28.6 | 662 | 25.6 | 1323 | 23.1 | 1321 |
| Presbyterian | 20.8 | 484 | 20.3 | 965 | 18.2 | 859 |
| Pentecost | 39.8 | 924 | 42.0 | 2245 | 46.6 | 2522 |
| Muslim | 10.7 | 337 | 12.1 | 507 | 12.1 | 571 |
| Living in rural | 81.0 | 2028 | 81.1 | 4358 | 82.6 | 4151 |

Notes: The study sample consists of adolescents aged 15–19 from Malawi Demographic and Health Surveys (MDHS): N = 2407 (2004 MDHS); N = 5040 (2010 MDHS) and N = 5273 (2015–16 MDHS). SD: standard deviation. [a] The median is mathematically equivalent to calculating an interpolated median between the completed ages 16 and 17, which lies between 45% and 55% cumulative frequency.

The bivariate analysis presented in Table 3 shows that overall, the proportion of adolescents who began childbearing or were pregnant was 34.1% in 2004, which declined to 25.6% in 2010 before increasing to 29.0% in 2015–16. Teenage childbearing was associated with early sex debut, marital status, education, employment status, wealth index, source of family planning messages, religion, ethnicity, place of residence and region. The prevalence of teenage childbearing was particularly high among those who had ever been married, had early sexual debut or had not attained education.

**Table 3.** Trends in prevalence of teenage childbearing from 2004 to 2015–16.

| Background Characteristics | 2004 MDHS | | | 2010 MDHS | | | 2015–16 MDHS | | |
| | Percent (C.I.) | n | $X^2$ | Percent (C.I.) | n | $X^2$ | Percent (C.I.) | n | $X^2$ |
|---|---|---|---|---|---|---|---|---|---|
| Age (years) | | | 617 | | | 1175 | | | 1097 |
| 15 | 3.20 (1.80–5.60) | 451 | | 3.50 (2.60–4.90) | 1246 | | 4.50 (3.30–6.20) | 1258 | |
| 16 | 11.5 (8.80–15.0) | 474 | | 12.6 (10.5–15.0) | 1171 | | 12.2 (9.90–14.9) | 971 | |
| 17 | 30.7 (26.0–36.0) | 420 | | 21.7 (18.4–25.4) | 944 | | 26.6 (23.2–30.2) | 941 | |
| 18 | 49.9 (44.8–55.0) | 578 | | 43.4 (39.0–47.9) | 883 | | 45.6 (42.0–49.2) | 1085 | |
| 19 | 67.9 (62.7–72.8) | 484 | | 63.5 (59.0–67.7) | 796 | | 59.2 (55.5–62.7) | 1018 | |
| Sexual debut | | | 647 | | | 1246 | | | 740 |
| Under 15 | 72.1 (68.0–75.9) | 706 | | 65.3 (61.7–68.8) | 1168 | | 58.0 (54.7–61.2) | 1303 | |
| 15 and over | 18.2 (15.8–20.8) | 1701 | | 13.7 (12.4–15.2) | 3872 | | 19.0 (17.5–20.6) | 3970 | |
| Marital status | | | 1444 | | | 3077 | | | 2863 |
| Never married | 6.40 (5.00–8.20) | 1504 | | 5.30 (4.50–6.30) | 3722 | | 9.00 (7.80–10.3) | 3909 | |
| Ever married | 82.7 (79.7–85.4) | 903 | | 82.8 (80.0–85.3) | 1318 | | 83.8 (81.1–86.1) | 1364 | |
| Highest education | | | 111 | | | 122 | | | 2809 |
| No education | 63.1 (53.7–71.7) | 134 | | 44.1 (34.8–53.9) | 140 | | 53.0 (43.3–62.4) | 126 | |
| Primary 1–4 | 41.8 (37.1–46.8) | 611 | | 33.5 (29.7–37.4) | 998 | | 35.3 (31.7–39.0) | 816 | |
| Primary 5–8 | 32.6 (29.2–36.2) | 1244 | | 25.4 (23.4–27.4) | 2843 | | 29.8 (27.7–32.1) | 3101 | |
| Secondary and higher | 19.0 (14.8–24.2) | 418 | | 15.6 (12.4–19.6) | 1059 | | 18.9 (16.0–22.3) | 1230 | |
| Employment status | | | 88 | | | 22 | | | 115 |
| Unemployed | 27.1 (24.2–30.3) | 1475 | | 23.4 (21.4–25.5) | 3097 | | 23.5 (21.8–25.4) | 3352 | |
| Employed | 45.9 (41.9–50.0) | 930 | | 29.4 (26.9–32.1) | 1932 | | 37.3 (34.4–40.2) | 1921 | |
| Wealth index | | | 100 | | | 107 | | | 116 |
| Poorest | 43.2 (37.8–48.8) | 423 | | 31.1 (27.2–35.2) | 957 | | 43.6 (39.5–47.8) | 863 | |
| Poorer | 46.9 (41.0–52.8) | 411 | | 31.1 (27.3–35.2) | 918 | | 34.8 (30.9–38.8) | 950 | |
| Middle | 35.8 (30.7–41.3) | 457 | | 30.2 (26.7–33.9) | 1004 | | 30.5 (27.0–34.2) | 979 | |
| Richer | 32.0 (27.2–37.2) | 537 | | 23.8 (20.4–27.5) | 1079 | | 24.7 (21.6–28.1) | 1071 | |
| Richest | 20.4 (16.1–25.3) | 579 | | 15.6 (12.7–19.1) | 1082 | | 15.3 (12.8–18.3) | 1410 | |

**Table 3.** *Cont.*

| Background Characteristics | 2004 MDHS | | | 2010 MDHS | | | 2015–16 MDHS | | |
|---|---|---|---|---|---|---|---|---|---|
| | Percent (C.I.) | n | $X^2$ | Percent (C.I.) | n | $X^2$ | Percent (C.I.) | n | $X^2$ |
| Source of family planning message | | | 41 | | | 64 | | | 239 |
| None | 33.2 (29.2–37.4) | 920 | | 23.6 (21.5–26.0) | 2321 | | 30.0 (28.1–32.0) | 3300 | |
| Radio | 39.9 (36.3–43.5) | 1050 | | 32.4 (29.6–35.4) | 1685 | | 32.2 (28.6–35.9) | 1163 | |
| Television | 21.0 (11.4–35.4) | 64 | | 23.9 (16.6–33.2) | 196 | | 17.8 (12.4–25.0) | 290 | |
| Newspaper | 22.9 (17.8–28.9) | 373 | | 18.9 (15.6–22.7) | 838 | | 21.5 (17.1–26.7) | 520 | |
| Ever used modern contraceptives | | | 190 | | | 747 | | | 37 |
| No | 30.2 (27.8–32.8) | 2222 | | 20.3 (18.8–21.9) | 4579 | | 20.7 (19.2–22.3) | 4471 | |
| Yes | 80.4 (73.3–86.0) | 185 | | 79.0 (73.5–83.5) | 461 | | 75.6 (71.2–79.5) | 802 | |
| Ethnicity | | | 23 | | | 33.6 | | | 993 |
| Chewa | 29.7 (25.6–34.1) | 757 | | 21.2 (18.6–24.1) | 1502 | | 25.4 (22.5–28.5) | 1646 | |
| Tumbuka | 33.5 (28.2–39.2) | 385 | | 25.7 (22.4–29.2) | 838 | | 27.5 (23.7–31.7) | 841 | |
| Ngoni | 35.0 (31.0–39.3) | 645 | | 27.4 (24.7–30.2) | 1490 | | 30.9 (28.1–33.9) | 1624 | |
| Yao | 44.7 (38.5–51.0) | 368 | | 31.4 (27.0–36.1) | 503 | | 34.7 (30.8–38.9) | 588 | |
| Other | 33.9 (26.8–41.8) | 252 | | 28.3 (23.8–33.4) | 707 | | 31.0 (26.9–35.5) | 574 | |
| Religion | | | 73 | | | 63 | | | 27 |
| Catholic | 31.6 (27.1–36.4) | 662 | | 22.4 (19.5–25.5) | 1323 | | 25.5 (22.6–28.6) | 1321 | |
| Presbyterian | 21.6 (17.5–26.2) | 484 | | 18.3 (15.5–21.4) | 965 | | 23.5 (19.7–27.8) | 859 | |
| Pentecost | 27.1 (19.1–36.8) | 924 | | 21.4 (16.5–27.2) | 2245 | | 22.7 (17.7–28.7) | 2522 | |
| Muslim | 42.3 (38.6–46.0) | 337 | | 31.1 (28.8–33.6) | 507 | | 33.5 (31.3–35.7) | 571 | |
| Residence | | | 22 | | | 16 | | | 51 |
| Urban | 24.8 (18.7–32.1) | 379 | | 20.5 (16.0–25.9) | 682 | | 21.3 (17.7–25.3) | 1122 | |
| Rural | 36.2 (33.6–39.0) | 2028 | | 26.8 (25.2–28.5) | 4358 | | 30.7 (28.9–32.5) | 4151 | |
| Total | 36.1 (31.5–36.7) | 2407 | | 25.6 (24.0–27.3) | 5040 | | 29.0 (27.4–30.7) | 5273 | |

Notes: C.I.: 95% confidence interval, $X^2$ = chi square.

Just as there are important differences in adolescent fertility by demographic and socioeconomic factors, Figure 1 shows that the prevalence of teenage childbearing varied significantly by district: teenage childbearing across districts ranged from 13 to 52 percent in 2004; from 16 to 38.4 percent in 2010 and from 15.6 to 41.1 percent in 2015–16. Of the 28 districts, 12 districts (i.e., Chitipa, Rumphi, Ntchisi, Likoma, Mzimba, Ntchisi, Lilongwe, Mchinji, Ntcheu, Machinga, Mwanza, Balaka and Neno) registered increased proportions between 2004 and 2015. The results show that between 2004 and 2010, 23 districts experienced a decline in teenage childbearing, while between 2010 and 2015, 22 districts experienced an increase in teenage childbearing. This alternative decline followed by increases in some districts worked to cancel the overall trend of decline in teenage fertility.

*3.2. Multilevel Results*

Table 4 presents individual and contextual predictors of teenage childbearing in Malawi for 2004, 2010 and 2015–16 surveys. The results show that three demographic variables (age, age at first sex and marital status) were significant in at least one of the years, which merits our approach for adjusting demographic and socioeconomic factors that influence teenage childbearing. Age of an adolescent and age at sexual debut have important implications on the sensitisation messages of girls abstaining from sex that could help reduce teenage childbearing, as well as the age at which adolescents receive appropriate messages that are embedded in the comprehensive sexuality education curricula in Malawi schools. Individuals' age and marital status consistently and significantly predicted the odds of teenage childbearing. However, the effect of age at first sex was strongest in 2004, with adolescents who delayed sex beyond 15 years having significantly lower odds than those who initiated sex before 15 years (O.R.: 0.5, $p < 0.001$).

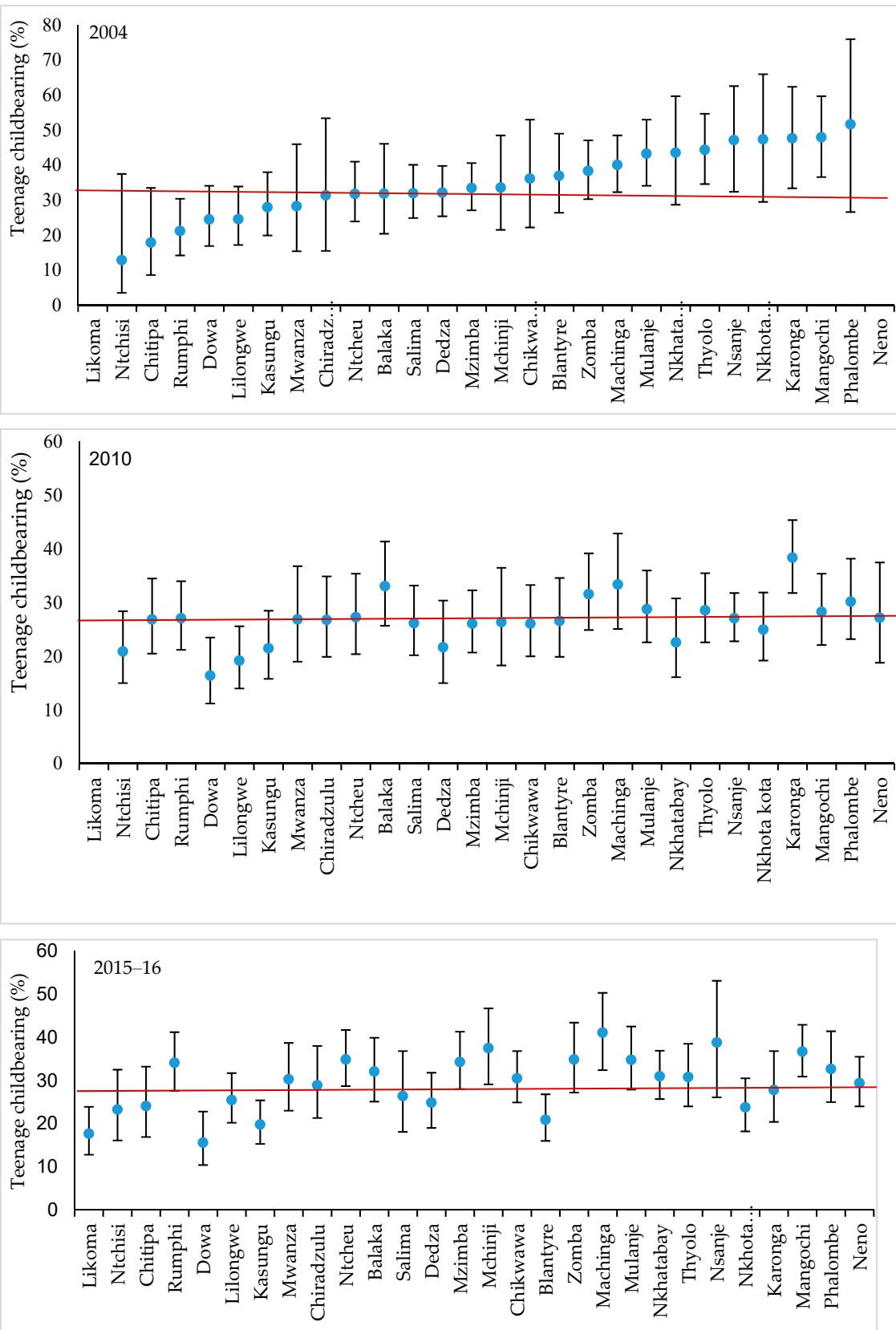

**Figure 1.** Prevalence of teenage childbearing by district from 2004 to 2015–16 MDHS.

**Table 4.** Odds ratios and 95 percent confidence intervals for demographic, socioeconomic and community variables associated with teenage childbearing in Malawi from 2004 to 2015–16.

| | 2004 MDHS | | | 2010 MDHS | | | 2015–16 MDHS | | |
|---|---|---|---|---|---|---|---|---|---|
| | Model 2 | Model 3 | Model 4 | Model 2 | Model 3 | Model 4 | Model 2 | Model 3 | Model 4 |
| **Age (Ref: 15)** | | | | | | | | | |
| 16 | 2.28 * (1.17–4.43) | | 2.38 * (1.20–4.70) | 2.44 *** (1.62–3.69) | | 2.41 *** (1.60–3.65) | 1.90 ** (1.29–2.80) | | 2.00 *** (1.35–2.98) |
| 17 | 3.97 *** (2.08–7.57) | | 4.14 *** (2.13–8.07) | 2.89 *** (1.90–4.39) | | 3.01 *** (1.97–4.59) | 4.20 *** (2.92–6.03) | | 4.36 *** (2.98–6.37) |
| 18 | 5.63 *** (2.99–10.6) | | 6.00 *** (3.12–11.5) | 5.08 *** (3.37–7.65) | | 5.38 *** (3.53–8.19) | 5.21 *** (3.63–7.47) | | 5.56 *** (3.81–8.13) |
| 19 | 11.6 *** (6.11–22.1) | | 12.33 *** (6.31–24.1) | 9.02 *** (5.9–13.7) | | 9.43 *** (6.08–14.63) | 6.97 *** (4.82–10.1) | | 7.56 *** (5.09–11.2) |
| **Early sex debut (Ref: under 15)** | | | | | | | | | |
| 15 and over | 0.51 *** (0.37–0.68) | | 0.49 *** (0.36–0.67) | 0.57 *** (0.44–0.72) | | 0.58 *** (0.45–0.75) | 0.81 (0.66–1.00) | | 0.79 * (0.63–0.98) |
| **Marital status (Ref: Never married)** | | | | | | | | | |
| Ever married | 35.2 *** (26.2–47.3) | | 28.9 *** (20.9–40.1) | 52.6 *** (41.6–66.5) | | 39.2 *** (30.3–50.7) | 37.5 *** (30.6–45.9) | | 26.3 *** (21.0–32.9) |
| **Education (Ref: No education)** | | | | | | | | | |
| Primary 1–4 | | 0.56 ** (0.37–0.84) | 1.07 (0.59–1.93) | | 0.84 (0.56–1.26) | 1.68 (0.91–3.11) | | 0.47 *** (0.31–0.72) | 0.99 (0.54–1.83) |
| Primary 5–8 | | 0.42 *** (0.28–0.63) | 1.18 (0.65–2.12) | | 0.50 *** (0.34–0.74) | 1.30 (0.72–2.37) | | 0.40 *** (0.27–0.60) | 0.84 (0.47–1.51) |
| Secondary and higher | | 0.31 *** (0.19–0.51) | 0.75 (0.37–1.53) | | 0.30 *** (0.19–0.47) | 0.72 (0.37–1.42) | | 0.27 *** (0.17–0.43) | 0.44 * (0.23–0.83) |
| **Employment status (Ref: Not employed)** | | | | | | | | | |
| Employed | | 1.88 *** (1.54–2.29) | 1.15 (0.85–1.56) | | 1.19 * (1.03–1.39) | 0.87 (0.69–1.09) | | 1.56 *** (1.35–1.81) | 0.99 (0.81–1.22) |
| **Wealth index (Ref: Poorest)** | | | | | | | | | |
| Poorer | | 1.37 * (1.02–1.84) | 1.07 (0.68–1.70) | | 0.91 (0.73–1.14) | 0.69 * (0.49–0.98) | | 0.70 ** (0.56–0.87) | 0.64 ** (0.47–0.89) |
| Middle | | 0.89 (0.66–1.20) | 1.00 (0.63–1.59) | | 0.92 (0.74–1.14) | 0.79 (0.56–1.12) | | 0.53 *** (0.42–0.66) | 0.79 (0.57–1.08) |
| Richer | | 0.69 * (0.51–0.93) | 0.75 (0.48–1.19) | | 0.67 *** (0.54–0.84) | 0.56 ** (0.39–0.80) | | 0.45 *** (0.36–0.57) | 0.78 (0.56–1.08) |
| Richest | | 0.44 *** (0.30–0.63) | 0.89 (0.50–1.58) | | 0.44 *** (0.33–0.58) | 0.62 * (0.41–0.95) | | 0.24 *** (0.18–0.32) | 0.40 *** (0.28–0.60) |
| **Source of family planning message (Ref: none)** | | | | | | | | | |
| Radio | | 1.59 *** (1.29–1.97) | 1.04 (0.75–1.43) | | 1.79 *** (1.52–2.10) | 1.36 * (1.06–1.75) | | 1.46 *** (1.23–1.74) | 1.06 (0.83–1.34) |
| Television | | 1.39 (0.71–2.69) | 1.00 (0.39–2.53) | | 1.30 (0.86–1.95) | 1.00 (0.55–1.82) | | 0.92 (0.63–1.34) | 0.95 (0.58–1.54) |
| Newspaper | | 1.00 (0.73–1.38) | 0.95 (0.60–1.52) | | 1.14 (0.90–1.44) | 0.94 (0.66–1.34) | | 0.85 (0.64–1.11) | 0.84 (0.59–1.19) |
| **Modern contraceptives** | | | | | | | | | |
| Yes | | 11.12 *** (7.41–16.7) | 5.51 *** (3.19–9.51) | | 19.01 *** (14.6–24.69) | 7.08 *** (4.83–10.4) | | 13.53 *** (11.2–16.4) | 6.35 *** (4.92–8.20) |
| **Ethnicity (Ref Chewa)** | | | | | | | | | |
| Tumbuka | | 1.18 (0.82–1.71) | 1.29 (0.74–2.23) | | 1.91 *** (1.47–2.49) | 1.34 (0.87–2.07) | | 1.56 ** (1.17–2.07) | 1.32 (0.94–1.86) |
| Ngoni | | 1.10 (0.81–1.51) | 1.02 (0.64–1.64) | | 1.61 *** (1.31–1.98) | 1.74 ** (1.22–2.47) | | 1.36 ** (1.11–1.67) | 1.87 *** (1.43–2.46) |
| Yao | | 0.88 (0.57–1.35) | 1.16 (0.61–2.20) | | 1.13 (0.78–1.64) | 0.78 (0.44–1.41) | | 1.08 (0.77–1.50) | 1.37 (0.90–2.10) |
| other | | 0.99 (0.67–1.46) | 1.09 (0.62–1.94) | | 1.35 * (1.04–1.77) | 1.19 (0.76–1.86) | | 1.70 *** (1.29–2.25) | 1.97 *** (1.37–2.83) |
| **Religion (Ref: Catholic)** | | | | | | | | | |
| Presbyterian | | 0.73 * (0.55–0.99) | 1.02 (0.66–1.58) | | 0.87 (0.69–1.10) | 0.87 (0.61–1.24) | | 1.12 (0.88–1.43) | 1.12 (0.81–1.53) |
| Pentecost | | 1.19 (0.94–1.50) | 1.16 (0.81–1.67) | | 1.19 (1.00–1.43) | 1.03 (0.78–1.36) | | 1.18 (0.99–1.42) | 0.94 (0.74–1.21) |
| Muslim | | 1.66 * (1.07–2.56) | 1.18 (0.61–2.26) | | 1.56 * (1.09–2.24) | 1.35 (0.77–2.37) | | 1.98 *** (1.41–2.79) | 1.70 * (1.09–2.65) |
| **Residence (Ref: Urban)** | | | | | | | | | |
| Rural | | 0.75 (0.53–1.06) | 1.17 (0.71–1.92) | | 0.94 (0.72–1.23) | 0.90 (0.61–1.33) | | 0.72 ** (0.57–0.91) | 0.63 ** (0.46–0.85) |
| *N* | 2407 | 2405 | 2405 | 5040 | 5029 | 5029 | 5273 | 5273 | 5273 |

Notes: Exponentiated coefficients; 95% confidence intervals in brackets; * $p < 0.05$, ** $p < 0.01$, *** $p < 0.001$; Model 1 (empty model); Model 2 (demographic factors); Model 3 (socioeconomic factors); Model 4 (demographic and socioeconomic factors).

In Model 3 of multilevel logistic regression, presented in Table 4, the following socioeconomic variables were significantly associated with teenage childbearing: education, ever used modern contraceptives, employment status, wealth, source of family planning message, religion, ethnicity and place of residence. Place of residence was not significant in 2004 and 2010; however, in 2015, adolescents who resided in rural areas had significantly reduced odds of experiencing a teenage birth (O.R.: 0.71, $p < 0.001$, C.I.: 0.57–0.90) compared to those living in urban areas. For Model 4 which controlled for both demographic and socioeconomic factors, age, marital status, use of modern contraceptives and employment were significantly associated with early motherhood in 2004. Additionally, in 2010, education was found to be associated with teenage childbearing, although there was no significant difference between adolescents who attained primary 1–4 and those with no education. Wealth, religion and ethnicity had at least one category that significantly influenced teenage childbearing. The model also examined the extent religion groups—Protestants, Catholics, Pentecost, Muslims—are positioned to provide protection against childbearing among their adolescent adherents. Female Muslim youths consistently exhibited higher incidence of teenage childbearing across the surveys. The majority of the variables described in the earlier two surveys turned out to be significant in 2015–16 when socioeconomic and demographic variables were accounted for. Furthermore, it is shown that additionally, delaying sex to age 15 and over (O.R.: 0.52, $p < 0.001$, C.I.: 0.40–0.68) and residing in rural areas (O.R.: 0.67, $p < 0.01$, C.I.: 0.49–0.92) were significant predictors in reducing teenage childbearing. In the analyses, age is one of the strongest predictors of teenage childbearing. After adjusting for the socioeconomic and demographic indicators, not only did it remain statistically strong, but also the value of the odds ratios increased over time. This is not surprising since increased age is associated with increased duration of exposure to the risk of teenage childbearing.

Having controlled for the individual level factors affecting teenage childbearing, the influence of district in which adolescents reside (shown in Table 5) indicates there still remains some unexplained variation at the district level. This means that there are other unobserved factors operating at district level which affected teenage childbearing. That is, the probability of experiencing teen childbearing among adolescents who share the same characteristics depends on the district in which they live. As already noted in Table 3, Model 1 (empty model) for each survey year, teenage childbearing varied significantly across districts for each survey year (2004 = O.R.: 0.52, $p < 0.001$; 2010 = O.R.: 0.35, $p < 0.001$; and 2016 = O.R.: 0.39, $p < 0.001$). The fraction of total variation in the data that is accounted for by between-group in districts, interclass correlation (ICC)—that is, among some adolescents who share the same attributes living in the same district—was 2.4%, 1.1% and 1.8% in 2004, 2010 and 2016, respectively (Table 4). The full model, after adjusting for the demographic and socioeconomic factors, shows that variations in teenage childbearing across districts remains statistically significant.

Moreover, the median odds ratio (MOR) confirmed that teenage childbearing was partly attributed to district level factors. If the MOR was equal to one, there would be no differences in the risk of teenage pregnancy across districts: a value of more than 1 indicates the presence of strong area differences suggesting that district is relevant for understanding variations of the probability of teenage childbearing. Table 5 also shows that the MOR for teenage childbearing was 1.31 in 2004, 1.20 in 2010 and 1.26 in 2015 in the empty models but decreased in the full model for 2004 and 2010 with the exception of 2010, where it increased. Nevertheless, this shows that there is significant clustering in teenage childbearing within districts.

**Table 5.** Measures of clustering.

|  | Model 1 (Null) | Model 2 | Model 3 | Model 4 |
|---|---|---|---|---|
|  | Estimate (S.E.) | Estimate (S.E.) | Estimate (S.E.) | Estimate (S.E.) |
| Constant (OR) |  |  |  |  |
| 2004 | 0.52 *** (0.04) | 0.03 *** (0.01) | 0.83 (0.25) | 0.02 *** (0.01) |
| 2010 | 0.35 *** (0.02) | 0.31 *** (0.01) | 0.32 *** (0.08) | 0.08 (0.04) |
| 2015 | 0.39 *** (0.02) | 0.19 (0.03) | 0.81 (0.21) | 0.05 *** (0.02) |
| District Level Variance |  |  |  |  |
| 2004 | 0.08 (0.04) | 0.16 (0.09) | 0.08 (0.05) | 0.16 (0.10) |
| 2010 | 0.04 (0.02) | 0.14 (0.06) | 0.02 (0.02) | 0.08 (0.05) |
| 2015 | 0.06 (0.02) | 0.02 (0.03) | 0.03 (0.02) | 0.02 (0.03) |
| Inter Cluster Correlation (ICC%) |  |  |  |  |
| 2004 | 2.42 | 4.71 | 2.34 | 4.63 |
| 2010 | 1.13 | 4.13 | 4.82 | 2.36 |
| 2015 | 1.76 | 0.70 | 0.8 | 0.66 |
| Median Odds Ratio (MOR) |  |  |  |  |
| 2004 | 1.31 | 1.47 | 1.31 | 1.47 |
| 2010 | 1.20 | 1.43 | 1.12 | 1.31 |
| 2015 | 1.26 | 1.16 | 1.16 | 1.15 |
| DIC (−2Log Likelihood) |  |  |  |  |
| 2004 | −1550.6 | −718.4 | −1345.2 | −688.6 |
| 2010 | −2888.0 | −1261.9 | −2391.1 | −948.0 |
| 2015 | −3128.8 | −1270.7 | −2464.0 | −1456.6 |

Notes: model 1 (empty model); model 2 (demographic factors); model 3 (socioeconomic factors); and model 4 (demographic and socioeconomic factors). S.E. Standard Errors; *** $p < 0.001$.

### 3.3. District Level Residuals

The estimates of district level residuals can be used to predict the district effect, after controlling for the sociodemographic characteristics of adolescents. Those with below-average teenage childbearing level would lie below the line and those with above-average teenage childbearing would lie above the line. The estimates for districts with average teenage childbearing level would lie on the horizontal line where the residual estimates are zero.

Overall, Figure 2 shows that some districts have higher than average teenage childbearing, while others have below-average levels. For example, 15 out of 28 districts have above average rates of teenage childbearing, while the rest have below average teen motherhood.

The caterpillar plots (Figure 2a,b) explore the variation in more depth, showing the estimated residuals for the second level variable. This is the case for Mangochi and Thyolo, where confidence intervals do not overlap the line. Figure 2b shows that controlling for all key variables, some districts may have higher rates of early marriage and early sex debut that also led to increased risk of adolescent fertility, hence making teenage childbearing ubiquitous in all the districts.

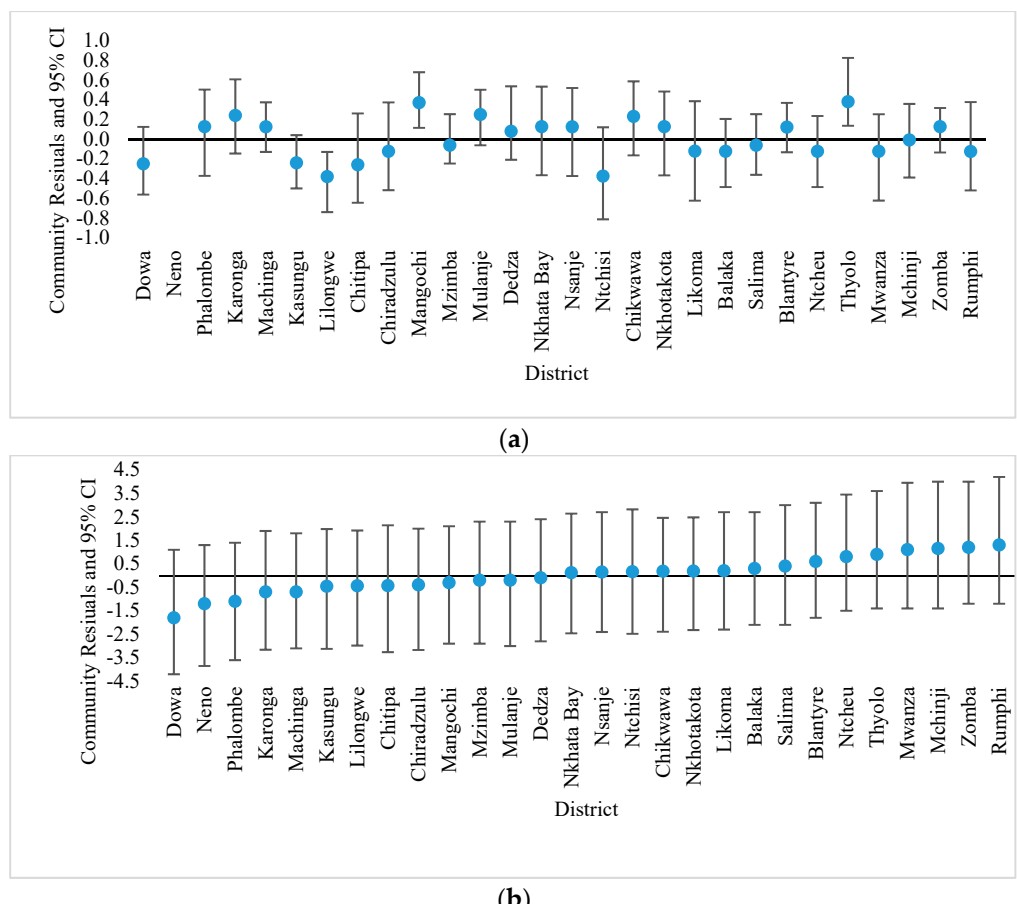

**Figure 2.** Caterpillar plot of random intercept predictions and 95% confidence intervals (CI) for null and full models 2015–16 MDHS.

## 4. Discussion

### 4.1. Effect of Socioeconomic Factors (Education, Wealth and Residence) on Teenage Childbearing

A multilevel analysis employed in this study sought to understand change over time in risk and protective factors influencing teenage childbearing in Malawi from 2004 to 2015. This is a critical step towards understanding ways that may reduce teenage childbearing prevalence and therefore help to direct resources towards interventions that are effective and efficient. The effects of education are in the expected direction. For example, attainment of secondary education was associated with reduced risk of adolescent childbearing. The more education a woman has, the more likely she is informed about ways of preventing early pregnancy, and the more aware she is of contraceptive options. In addition, it has been hypothesised that educated adolescents have positive prospects for success in future (Brown et al. 2002) and education becomes a protective factor against early childbearing and unintended pregnancy (Chalasani et al. 2012; Girma and Paton 2015; Ayele et al. 2018).

The analyses have revealed a link not only between teenage pregnancy and level of education but also between teenage pregnancy and wealth (NSO and ICF 2017). The study found that adolescents who belonged to highest quintiles have reduced odds of teenage childbearing. Earlier findings suggest that adolescents who belong to lower socioeconomic status are more likely to begin childbearing or become pregnant (Wado et al. 2019). This may suggest that teenage pregnancies and early marriages associated with low levels of education might be adaptive responses to economic conditions (Kassa et al. 2018).

The multivariate results show that with respect to childbearing, the urban–rural setting was not significant in 2004 and 2010, suggesting that the other variables explain the rural–urban difference. While the 2015 findings contradict patterns observed in the bivariate relationship which indicate that adolescent childbearing was higher in the rural areas

compared to the urban, in the multivariate analysis, residing in urban areas is associated with an increased risk of teenage childbearing. Urban life disrupts traditional family roles that attach stigmas to premarital sexual experimentation patterns and diminishes parental control over adolescents (Brown et al. 2002).

### 4.2. Influence of Sociocultural Factors on Teenage Fertility

In the case of religion, the findings showed that after controlling for socioeconomic factors (Model 3) in all survey years, female Muslim adolescents had increased odds of childbearing compared to their Catholic counterparts. Lack of protective factors among female Muslim adolescents established in this study is in agreement with Munthali et al. (2006), who proposed that religion still exerts influence on sexual and reproductive health behaviour of adolescents in the country. One mechanism that exposes adolescents is the practice of initiation among the Yao—a predominantly Muslim ethnic group found in Salima, Mangochi and Machinga districts of Malawi. An anthropological study in Mangochi found that initiates are encouraged to practice sex after initiation often with older adults to mark transition to adulthood (Pot 2019). Given that the practice sanctions early sexual debut, adolescents who engage in sexual experimentation are at an increased risk of sexually transmitted diseases, including HIV/AIDs because sexual activity occurs with limited knowledge of sexual reproductive health information and services (YFHS 2014). Elsewhere, Heaton (2011) confirms that Muslims tend to have earlier marriage and to be more tolerant of polygyny.

It suffices to mention that programmatic factors appear to correlate with the probability of teenage childbearing. In particular, adolescents who listen to radio have increased odds of experiencing childbearing. The bivariate analyses showed that there has been a steady decline in access to family planning messages through radio, newspaper and television, and the proportion that did not receive family planning messages from the three sources increased. This may reflect a weakening of family planning programmes over time or may suggest that the programmes aired on radios do not carry the preventive messages. One plausible explanation for the reverse relationship could be that with an increase in youth friendly health services, adolescents saw less of a need to listen to family planning messages at home in the presence of adults.

Other studies found that outreach programmes are generally broadly based and may not reach adolescents especially with pressing SRH needs (Erulkar et al. 2006; Chandra-Mouli et al. 2015). This standpoint is further strengthened because once demographic factors (Model 4) are accounted for, the significant positive influence of the programmatic variables on teenage childbearing disappears. A recent qualitative study found that interventions focusing on adolescents in Malawi have not yielded the intended results as programs benefitted a minority of the target population who tended to be male, older and more educated (Self et al. 2018). It is also likely that media messages may be designed for less conservative urban adolescents, yet the majority (81%) of adolescents reside in the rural areas where cultures are conservative.

The positive relationship between employment and teenage childbearing found in this study supports Ayele et al. (2018) and Birhanu et al. (2019) findings which suggest that teenage pregnancy is likely to lead to school drop-out leading to adolescents being employed in low-paying jobs in order to achieve economic independence from their own parents and meet the needs of their children. With fewer skills and other constraints that often accompany early pregnancy, girls are likely to find it difficult to enter the paid labour force (Temmerman 2017). The findings show that correlation between use of modern contraceptives and teenage childbearing is less clear: use of modern contraceptives does not provide the expected protective effect on teenage childbearing. This finding is similar to studies conducted in three Latin America countries (Colombia, the Dominican Republic and Honduras) which found that use of modern contraceptives among adolescents was positively correlated with teenage pregnancy (Azevedo et al. 2012; Arceo-Gómez and Campos-Vazque 2014). First, it might be the case of intense sexual activity and use of

temporary contraceptive methods, which offer little protection against pregnancy under the circumstances. Second, the girls in most cases are informed about contraceptives and start using them after pregnancy or childbearing to prevent any more unintended pregnancies, which may account for a high prevalence of contraceptive use amongst teenagers with children. Third, although contraception knowledge is high among young women and some know where to obtain it, this does not necessarily translate into use due to economic and physical barriers in accessing contraception, which may register high but inconsistent use of contraception (Meekers et al. 2001; Rosenberg et al. 2018).

## 5. Conclusions

This study has demonstrated that teenage fertility remains high in Malawi, which is primarily associated with early sexual activity and early marriage—factors that have hardly changed over time. Teenage childbearing remains, not only a societal concern, but presents a challenging transition to adulthood since childbearing can carry health, economic and social costs for mothers and their children. More attention needs to be given to tackling childbearing in the country, and a good place to start is the teenage population. A focus on teenagers is warranted because an increasing number of teens are becoming sexually active at an early age without using any protection, thereby putting themselves at risk of pregnancy and, in some cases, sexually transmitted infections including HIV and AIDS. These efforts include youth driven interventions with emphasis on sex education for in-school and out-of-school adolescents. Media campaigns on sexual and reproduction health on radio, television and newspapers should also be intensified to reach communities in remote and hard-to-reach areas so that adolescents should be kept abreast of appropriate behaviour change messages.

Further, there should be greater emphasis put on girls' rights to opportunities equal to those of boys no matter how sexual and reproductive health education. Developing a community-based approach which utilizes school sex education integrated with parent, church and community groups should be considered to increase teenage knowledge of contraception. The protective effect of educational attainment underscores the importance of empowering girls to enable them to make appropriate decisions leading to favourable outcomes.

*Limitation*

There are some important limitations to this study. First, certain variables in the study were used as proxy measures to assess the sociocultural influence of teenage childbearing. For example, religion and ethnicity may not have clearly defined the relationship in the process, which may have underestimated their contribution to the study. This may also have created challenges in estimating level 1 variance of the binomial outcome. This is because, the variance of a binomial is a direct function of the mean, which means that by definition, that level 1 variances and means are not completely independent. The interclass correlation coefficient for within districts is small, which may suggest that those large variations are located at the lower level compared to the variation between teenage childbearing within districts (level 2). This was accounted for by additionally using MOR to test for within-district effects in all the models, and the results showed that there was change associated with different districts in teenage childbearing. It would have been helpful to include contextual district-level measures to better understand important district-level predictors of teenage childbearing in Malawi. However, this was not possible since MDHS data have limited district level measures.

Second, the relationship between the level of education and other socioeconomic factors influencing childbearing is rather complex and presents challenges in causality issues. Although these variables were introduced one at a time in the models, as Eloundou-Enyegue and Stokes (2004) argue, poor socioeconomic environment may lead to adolescent girls engaging in premarital sex, resulting into pregnancy which could in turn lead to school dropout. Adolescents from poor households are more likely to drop out of school

which puts them at a higher risk of early marriage and childbearing. Finally, since the outcome variable is restricted to adolescents who gave birth during the five years preceding the survey or were pregnant at the time of interview, the socioeconomic characteristics may not correspond to the status of the adolescent at the time of pregnancy or childbirth.

Third, the present analyses correspond to the data that were collected in 2015. At the time of publication, there have not been any data that collect information on teenage childbearing that could provide more recent changes in teenage childbearing patterns.

**Author Contributions:** Conceptualization, J.C. and M.M.; methodology, J.C.; software, J.C.; validation, J.C., M.M. and G.L.; formal analysis, J.C.; investigation, M.M.; resources, G.L.; data curation, J.C.; writing—original draft preparation, J.C.; writing—review and editing, M.M.; visualization, M.M.; supervision, M.M.; project administration, G.L.; funding acquisition, G.L. All authors have read and agreed to the published version of the manuscript.

**Funding:** This research was funded by GLOBAL CHALLENGES RESEARCH FUND, grant number QR–GCRF (Research England) pump-priming fund and the APC was funded by MDPI.

**Institutional Review Board Statement:** DHS surveys have been reviewed and approved by ICF Institutional Review Board (IRB) available from https://dhsprogram.com/methodology/Protecting-the-Privacy-of-DHS-Survey-Respondents.cfm.

**Informed Consent Statement:** The study used secondary data and informed consent was obtained by the Demographic and Health Surveys Program at the time of data collection available from https://dhsprogram.com/methodology/Protecting-the-Privacy-of-DHS-Survey-Respondents.cfm.

**Data Availability Statement:** All data are available from https://dhsprogram.com/data/availabledatasets.cfm.

**Conflicts of Interest:** The authors declare no conflict of interest.

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
