# Peer review of "A Multilevel Analysis of Risk and Protective Factors for Adolescent Childbearing in Malawi"

_socsci, doi:10.3390/socsci10080303_

Round 1

Reviewer 1 Report

Thank you for the opportunity to review this manuscript.

I suggest that the authors can reduce the complexity and the length of the Introduction Section, focusing it only on the main points of this study without other irrelevant information. 

About the description of variables, the tables are not clearly legible. I believe that some variables can be included in the main text of the manuscript.

Are the authors sure that a detailed description of multilevel regression analysis is necessary? I don't think so. 

I believe that this paper deal with an important topic for the Malawi society, but it is necessary to make it more fluent and accessible to a wider audience.

Author Response

Dear Reviewer,

We would like to thank you for identifying the weaknesses in our paper and providing us the opportunity to strengthen our research prior to publication. We have addressed the questions and comments (in ‘red’ colour), and indicated the changes in the manuscript (track changed) in the revised manuscript.

Kind regards

Authors

Reviewer 2 Report

COMMENTS TO AUTHORS

Thank you for this very interesting and important paper that I enjoyed reading. It addresses an important topic and one that continues to require attention.  

The paper is written well and structured appropriately. I have only minor recommendations that I hope are helpful.

INTRODUCTION

I have no specific comments here – this section provides a valuable overview of the topic of teenage pregnancy, and the antecedents. There’s appropriate emphasis given to structural factors that play a role. The need for a closer look at the Malawian situation is justified in the text.

MATERIALS AND METHODS

The design and analysis are well-described, and as this is using secondary data it’s useful to see when primary data were collected.

The selected variable and analytical approaches seem appropriate to examine the data logically.

RESULTS

These are provided in suitable detail and are interesting, especially differences between the data points. It’s encouraging to see that early sexual debut has fallen between 2004 and 2015-16. Geographical data are also valuable to see, as is declared religion.

DISCUSSION

This section draws appropriately on the data and provides additional citations to place the findings in what is currently known. You highlight the key findings, and possible causes of data trends.

One minor issue I have with the paper is that it focuses almost entirely on structural factors that may influence teenage pregnancy. This is not a bad thing in itself of course (this element is often missed). But I wonder if, in the recommendations for solutions, a note or two can be added about ways to increase the agency of adolescent girls/young women (or at least acknowledging that this is important to address). How can they be more empowered? Taking a binary approach (addressing structural issues and increasing agency) can have a significant positive impact.

CONCLUSION

I note that you include limitations, which is helpful. I would also recommend adding that data are at least 6 years old; the situation may have changed by the time this paper is published.

REVIEWER RECOMMENDATIONS

  1. Add a limitation that data are at least 6 years old – acknowledge this, and perhaps comment on what the current situation may be.
  2. Add a note about the importance of empowering adolescent girls/young women as well as addressing structural issues to reduce teenage pregnancy.

Author Response

Dear Reviewer,

We very much appreciate the invaluable comments from the reviewers. Each of the points raised has been given careful consideration as shown in our response below (in ‘red’ colour) after each point.  

Please see the attachements.

Kind regards

Authors
